# Age-Related Changes in the Matrisome of the Mouse Skeletal Muscle

**DOI:** 10.3390/ijms221910564

**Published:** 2021-09-29

**Authors:** Francesco Demetrio Lofaro, Barbara Cisterna, Maria Assunta Lacavalla, Federico Boschi, Manuela Malatesta, Daniela Quaglino, Carlo Zancanaro, Federica Boraldi

**Affiliations:** 1Department of Life Sciences, University of Modena and Reggio Emilia, I-44100 Modena, Italy; francescodemetrio.lofaro@unimore.it (F.D.L.); daniela.quaglino@unimore.it (D.Q.); 2Department of Neurological and Movement Sciences, University of Verona, I-37100 Verona, Italy; barbara.cisterna@univr.it (B.C.); mariaassunta.lacavalla@univr.it (M.A.L.); manuela.malatesta@univr.it (M.M.); 3Department of Computer Science, University of Verona, I-37100 Verona, Italy; federico.boschi@univr.it

**Keywords:** aging, extracellular matrix, proteomics, ultrastructure, immunohistochemistry, sarcopenia

## Abstract

Aging is characterized by a progressive decline of skeletal muscle (SM) mass and strength which may lead to sarcopenia in older persons. To date, a limited number of studies have been performed in the old SM looking at the whole, complex network of the extracellular matrix (i.e., matrisome) and its aging-associated changes. In this study, skeletal muscle proteins were isolated from whole gastrocnemius muscles of adult (12 mo.) and old (24 mo.) mice using three sequential extractions, each one analyzed by liquid chromatography with tandem mass spectrometry. Muscle sections were investigated using fluorescence- and transmission electron microscopy. This study provided the first characterization of the matrisome in the old SM demonstrating several statistically significantly increased matrisome proteins in the old vs. adult SM. Several proteomic findings were confirmed and expanded by morphological data. The current findings shed new light on the mutually cooperative interplay between cells and the extracellular environment in the aging SM. These data open the door for a better understanding of the mechanisms modulating myocellular behavior in aging (e.g., by altering mechano-sensing stimuli as well as signaling pathways) and their contribution to age-dependent muscle dysfunction.

## 1. Introduction

Skeletal muscle (SM) is necessary for locomotion, but it also plays important roles in several physiological processes such as bone homeostasis, thermogenesis, and metabolism of amino acids, glucose, and lipids [1,2]. The progressive loss of muscle strength and mass, alterations in tissue composition, and increasing denervation can lead to the development of sarcopenia, a pathological condition which generally occurs in aging but can also occur at a young age [3,4]. Sarcopenia contributes to a lower quality of life since, for example, the risk of falls with consequent fractures and loss of independence increases and may contribute to the development of type II diabetes and of metabolic syndrome [5].

Over the past decades, many studies have shown that sarcopenia is a multifactorial process [6] involving, among other factors: (i) reduction in the size and the number of myofibers [7]; (ii) satellite cell (SC) exhaustion and altered immune and muscle cell cross-talk (necessary for muscle proliferative and regenerative capacity) [8]; (iii) mitochondria dysfunction causing inefficient energy production [9]; (iv) alterations in insulin-like growth factor 1, Notch, and Wnt/beta-catenin signaling pathways [3]; (v) increase of oxidative stress [10]; and (vi) dysregulation of protein synthesis and degradation [11].

The extracellular matrix (ECM) is composed of different groups of macromolecules such as collagens, non-collagenous glycoproteins (e.g., laminin, tenascin, and fibronectin), glycosaminoglycans (e.g., heparan sulfate), and proteoglycans (e.g., biglycan and lumican) [12]. These macromolecules bind to each other and to cells through integrins, sarcoglycan complex, and dystroglycan to form an intricate network sending biochemical signals to myofibres. ECM regulates several cell functions (e.g., growth, differentiation, and migration) [13] and represents the structural and functional support for muscle fibers, vessels, and nerves, playing a role in the transmission of contractile forces [14], as well as muscle development [15], growth, and repair [16]. ECM is affected by the aging process in terms of turnover and ratio of specific components, the balance between synthesis and degradation of components, modifications in cellular behavior through altered cell-matrix interactions, and changes in mechano-sensing pathways. In age-dependent sarcopenia, changes in ECM architecture and composition and fibrosis [17] are believed to reduce the regenerative capacity of SM and to negatively influence the proliferation and differentiation capability of SC [12,18,19,20]. The ECM remodeling in sarcopenia has also been linked to mitochondrial deterioration [9,19]. However, until now, a limited number of investigations have been performed in the aging SM looking at the matrisome as the whole complex network of ECM molecules.

In recent years, thanks to the development of mass spectrometry (MS)-based high-throughput proteomic techniques, large-scale protein characterization is less challenging. Proteomic approaches have been successfully applied to SM in different experimental models [21,22,23,24]. However, to date, there is little information available on ECM proteins of SM [25] and their aging-associated changes.

The identification and quantification of ECM components and their interactions are essential steps to understand the role of the matrisome in sarcopenia. In this context, we decided to investigate the SM ECM in old compared to adult mice. We are aware that it would be of interest to analyze life-long changes, and thus also include muscles from young animals. However, it has to be underlined that most changes in muscle protein expression are known to take place after middle age [24].

The morphology of old SM has been already characterized in previous work from our laboratory highlighting several typical features of the sarcopenic condition. A significantly smaller myofiber cross-sectional area was observed in the SM of old mice [26,27]. Although the general cytological organization of the old myofiber was not grossly altered, morphometrical studies highlighted several age-related modifications (e.g., accumulation of larger inter-myofibrillar and sub-sarcolemmal mitochondria, larger myonuclei with increased condensed chromatin, impairment of RNA maturation/export pathways, decreased amount of active satellite cells) [28,29,30]. In this work, the gastrocnemius muscle was selected for analysis since it is prevalently composed of fast-twitch fibers [27], which are especially affected by atrophy during aging.

Therefore, to shed light on age-related changes in the matrisome of the gastrocnemius muscle, we combined a proteomic approach (i.e., liquid chromatography (LC)-MS/MS and bioinformatic analyses) with morphological and morphometrical evaluations of sections observed by fluorescence and transmission electron microscopy.

Results provided novel characterization of the aging matrisome. Higher amounts of several ECM proteins in old vs. adult muscle were demonstrated, thereby shedding new light on the mutually cooperative interplay between cells and the extracellular environment. This study opens the door to a better understanding of the mechanisms modulating myocellular behavior (e.g., alteration of mechano-sensing stimuli and/or signaling pathways).

## 2. Results

### 2.1. Identification of Proteins in the Gastrocnemius Muscle

In this study, samples of gastrocnemius muscle were subjected to three sequential extractions, each one being analyzed by LC-MS/MS. In particular, phosphate buffer saline (PBS) was used to solubilize hydrophilic proteins (PBS extract), and then the insoluble part was treated with a combination of urea (U) and thiourea (T) to extract hydrophobic molecules such as membrane proteins, myofibrillar, and part of ECM proteins (U/T extract) [31]. Finally, the remaining insoluble fraction was treated with guanidine-HCl (GuHCl extract), one of the most efficient chaotropic agents which is known to extract poorly soluble, heavily cross-linked proteins, and proteoglycans. This approach has been reported to improve protein extraction from ECM-rich tissues (e.g., cartilage, tendon) [32,33] (Figure 1a).

This strategy allowed us to identify, with at least 1 unique peptide, 2134 different proteins (Appendix A) of which 474 were identified in all 3 extracts, while 588, 364, and 140 proteins were detected in the PBS, U/T, and GuHCl extracts, respectively (Figure 1b).

To reveal the ECM composition of the gastrocnemius muscle, the 2134 identified proteins were further analyzed by interrogating the MatrisomeDB, a database that includes all structural ECM components and proteins that may directly or indirectly interact with the ECM [34]. The matrisome is constituted by a “core matrisome”, comprising collagens, proteoglycans, and glycoproteins, and by “matrisome-associated proteins”, which include secreted factors, ECM-affiliated proteins, and ECM regulators [35]. As shown in Table 1, 124 proteins were found to be part of ECM; in particular, 14, 38, and 10 proteins belong to the collagen, glycoprotein, and proteoglycan category, respectively. Sixty-two proteins constituted the “matrisome-associated proteins” (i.e., 15, 37, and 10 proteins representing ECM-affiliated proteins, ECM regulators, and secreted factors, respectively). 43% (54/124) of proteins were found only after U/T and GuHCl extract of which 28 polypeptides were identified in both extracts (i.e., U/T and GuHCl) and 13 only with U/T or with GuHCl buffer.

### 2.2. Matrisome Quantification: Old SM vs. Adult SM

To compare the relative abundance of matrisome proteins between adult and old muscle, we performed a label-free quantification, which was based on the measure of precursor ion intensities. Even though protein quantification can be performed on proteins identified with a single peptide [36], we preferred to quantify matrisome proteins which were identified with at least two peptides (i.e., 91/124 proteins) (for more details see material and methods) (Appendix A), as only one peptide can be incorrectly quantified across LC-MS runs [37].

Table 2 lists only proteins that changed significantly between old and adult muscles (i.e., 18, 21, and 3 in PBS, U/T, and GuHCl extracts, respectively). Except for AnnexinA6, whose amount was significantly lower in aging SM, all other listed proteins were statistically significantly more abundant in old vs. adult SM. No statistically significant changes were found for proteins belonging to the category of secreted factors.

### 2.3. Ultrastructural Morphology and Morphometrical Evaluation

To correlate matrisome changes with structural features, gastrocnemius muscles from adult and old mice were collected and processed for morphological and morphometrical analyses using transmission electron microscopy.

The general organization of myofibers in adult and old mice is shown in Figure 2a,c, confirming previous works from our laboratory [28,29]. In both age groups, the endomysium was comprised of a network of collagen fibrils connected to the basement membrane, which was covering the surface of skeletal muscle cells an electrodense sheath (Figure 2b,d).

Morphometrical evaluation of the endomysium thickness performed measuring the distance between the sarcolemma of two adjacent longitudinally sectioned myofibers, revealed no statistically significant difference in adult vs. old mice (0.685 ± 0.041 and 0.611 ± 0.048; *p* = 0.25, Figure 2e). Instead, the thickness of the basement membrane was significantly higher in old vs. adult mice (29.72 ± 0.61 nm vs. 41.88 ± 1.07 nm; *p* < 0.001; Figure 2f), in agreement with the increased amount of basement membrane’s components identified by proteomic and immunofluorescence analyses (vide infra).

The extracellular matrix of the perimysium was organized in collagen bundles of different sizes and orientations (Figure 3a,c). In comparison with adult animals, old mice perimysium presented statistically significantly larger collagen bundles (759.16 ± 52.05 nm vs. 1461.66 ± 75.2 nm; *p* < 0.001; Figure 3e), which were also more linear (linearity index: 1.05 ± 0.01 vs. 1.03 ± 0.03; *p* = 0.03; Figure 3f). Morphometrical evaluation of the collagen fibrils of the perimysium (Figure 3b,d) revealed no statistically significant difference in size in adult vs. old mice (21.0 ± 0.65 nm and 21.4 ± 0.50 nm; *p* = 0.62, Figure 3g). The distance between collagen fibrils was statistically significantly higher in old vs. adult mice (9.26 ± 0.92 nm and 5.84.45 ± 0.56 nm; *p* = 0.006; Figure 3h).

### 2.4. Evaluation of ECM Components by Immunofluorescence

Immunofluorescence was performed on gastrocnemius cryosections from adult and old mice using specific antibodies directed against the ECM components collagen type VI (as representative interconnection factor between basement membrane and fibrillar collagen) and laminin (as constituent of the basement membrane) (Figure 4). Differences in the immunolabelling distribution were present in the two age groups. In particular, the immunostaining for both collagen type VI (Figure 4a,d) and laminin (Figure 4b,e) appeared more abundant in old vs. adult mice. Qualitative observations were confirmed by quantifying the area covered by fluorescence-positive pixels through a MATLAB routine. Both type VI collagen (0.1368 ± 0.0090 vs. 0.0621 ± 0.0097, *p* < 0.001, Figure 4g) and laminin (0.1728 ± 0.0084 vs. 0.0707 ± 0.0071; *p* < 0.001, Figure 4h) were statistically significantly increased in old SM.

Immunolabelling for type I collagen (as representative of fibrillar collagens) (Figure 5) revealed qualitatively different staining in the two groups of mice. Quantification of the area covered by fluorescence-positive pixels for type I collagen showed no statistically significant difference in the two age groups (adult, 0.0359 ± 0.0086; old, 0.0514 ± 0.0113; *p* = 0.275).

## 3. Discussion

In aging, SM can suffer from low muscle strength and low muscle quality and quantity which may lead to sarcopenia [4].

To characterize and to understand the processes of sarcopenia, many studies using diverse techniques (e.g., computerized tomography, magnetic resonance imaging, electron microscopy, transcriptomic analyses) have mainly focused on changes of the SM mechanical properties and/or alterations of myofibers and SC [38,39]. Moreover, thanks to the development of MS-based proteomics, the intracellular proteins of the aged SM were investigated [40,41]. However, little emphasis has been given to the matrisome, despite growing evidence indicating the ECM as an essential player in SM function. In fact, the matrisome not only constitutes the architectural scaffold for myofibers [42,43], but it is also a reservoir for signaling molecules (e.g., cytokines, chemokines, and growth factors) responding to different stimuli and to stress to maintain cellular homeostasis. Actually, the matrisome is a dynamic compartment mediating outside-in signaling and vice-versa between cells and the surrounding environment [44]. Therefore, in this study, we focused on the matrisome of the whole gastrocnemius muscle by applying a sequential extraction procedure in combination with LC-MS/MS analysis to reveal changes occurring with aging. It is worth remembering that, to date, there is no protocol and/or proteomic technique capable of detecting the entire complexity of the proteome, so our strategy was a compromise between sensitivity and quantitative accuracy. We were able to identify many ECM components, some of them (e.g., collagens, proteoglycans, glycoproteins) being only extracted after strong solubilization conditions such as urea/thiourea and/or guanidinium-HCl.

Our proteomic data demonstrated that changes in matrisome proteins mostly consisted of increased protein abundance in old vs. adult skeletal muscles.

### 3.1. Alteration of Core Matrisome in Aged vs. Adult Mice

Among the “core matrisome” **collagen** category, we revealed a significant age-dependent increase of collagen type IV and VI, which was confirmed by quantitative immunofluorescence.

Collagen type VI is present in the epimysial, perimysial, and endomysial interstitium playing a key role in maintaining the SM functional integrity as it represents, together with fibronectin and proteoglycans, the major constituent of the “niche”, where the balance between differentiation, self-renewal, and maintenance of muscle regeneration capacity takes place [20,45]. Collagen type VI is typically organized as a heterotrimer of α1, α2, and α3 chains, but, recently, three new α chains have been described [46]. Since the α chains have a different length and we have demonstrated an increased abundance of only α1 and α2 chain, it could be suggested that an altered ratio among the various collagen type VI chains affects the stability and the supramolecular assembly of the beaded filaments, altering the role of type VI collagen as an anchoring component of basement membranes through the interactions with collagen type IV, biglycan, fibronectin, sarcolemma integrins, and cell surface proteoglycans [46]. Interestingly, thickening of the immunofluorescence staining area for type VI collagen occurred in the gastrocnemius of old mice, further supporting an age-associated alteration of the organization of the anchoring system.

Collagen type IV is a structural molecule of the basement membrane, and its concentration appeared to increase in old muscle, in agreement with previous observations, [47] although with differences depending on muscle type. Collagen type IV accumulation has been shown to drive out SC from their niche, thus potentially contributing to the satellite cell reduction observed in aging [29,48,49].

It is to be noted that, despite the increased amount of type IV and VI collagen contributing to the increased thickness of basement membranes, the endomysium thickness was not changed in adult and old mice. Accordingly, no overall age-associated hypertrophy of the ECM component was previously demonstrated in mouse skeletal muscle [50]. Moreover, in the present study, the amount of fibrillar collagens (i.e., collagen type I, III, and V) in the mouse gastrocnemius muscle did not change with aging. Data reported in the literature are very heterogeneous [51,52,53,54], but the loss of muscle performance has been generally related to fibrosis, although, by transcriptomic analysis, collagen genes (i.e., collagen type I and III) appeared significantly downregulated in aging [55]. The discrepancy among different studies may be related to the type or specific portion of muscles analyzed, to sample preparation, and/or to methods applied to quantify collagen (e.g., hydroxyproline measured by HPLC, collagen staining, transcriptomic analysis, proteomic analysis). Moreover, collagen accumulation may be due to excessive production, the altered balance between degradative enzymes and their inhibitors, a combination of the altered ratio between synthesis and degradation, or may be the result of post-translational modification and glycation crosslinking, thus increasing insolubility and lowering the ability of proteolytic enzymes to provide an efficient turnover [56].

Even if we did not observe statistically significant differences in the amount of collagen type I by LC-MS/MS and immunofluorescence, the fibrillar collagen in the perimysium (predominantly collagen type I) [57] of old animals showed a more linear and loosely organized distribution with an increased distance between collagen fibrils that may likely allow the interposition of other ECM constituents. Interestingly, a statistically significant decrease in collagen fiber “tortuosity” [19] and the accumulation of extensively cross-linked collagen together with the reduced size of most myofibers [26,27] have been advocated as a cause of increased muscle stiffness [49,50]. Muscle stiffness has been recently reported to increase, together with ECM amount, in human skeletal muscle of aged vs. young subjects [58]. Accordingly, muscle stiffness is considered a typical hallmark of muscle aging, [50,51,52,59] as well as the decreased compliance of myofibers in response to tensile loading [19].

Other components of the matrisome are **glycoproteins** such as laminin, fibronectin, and nidogen, which were detected in higher amounts in old compared to adult SM. Since these glycoproteins act as a bridge between collagen type IV and the sarcolemma of muscle fibers [60], their increased amount is consistent with the higher thickness of the basement membrane observed in old animals. Moreover, the increase of laminin during aging can modify the ability of the basement membrane to store and release growth factors and other bioactive molecules required to form the SC niche [61] and to activate SC, thus supporting the previous finding of SC reduced activation potential in old mice [29]. In aged muscle, the shift from functional myofiber repair towards increased extracellular matrix deposition is also associated with changes in the micro-environment and the interactions between ECM glycoproteins as fibronectin, periostin, and tenascin. Interestingly, all of these molecules appeared to increase in the old gastrocnemius muscle.

Fibronectin, for instance, is one of the most widespread glycoproteins of the ECM, playing a role in various processes such as cellular adhesion, spreading and migration, as well as cellular development and differentiation. The increased abundance of fibronectin in old gastrocnemius is in agreement with very recent data demonstrating an association between high fibronectin levels in the aged gastrocnemius muscle, the reduced muscular strength, and myogenic regeneration and differentiation [62].

Periostin is a member of the TGF-beta family of proteins with a putative role associated with pathologic fibrotic events [63]. It is present in the endomysial space and functions upstream and downstream of TGF-beta [64]. Its expression is low in adult tissues, but it is strongly induced and secreted after injury or in dystrophic skeletal muscle [65].

Tenascin is present in all musculoskeletal regions in which high mechanical forces are transmitted, is upregulated in the damage/repair cycle [66] being produced in response to degeneration and the release of growth factors as TGF-beta, and was found to accumulate in the endomysium predominantly in the vicinity of necrotic and regenerating myofibers [67]. As it was also observed in association with atrophy and with age-induced atrophy, it may represent a possible cofactor in the etiology of sarcopenia [68,69].

Interestingly, thrombospondin (TSP1) is also an ECM molecule that was observed to be markedly increased in the aging SM. It modulates cell function by binding to matrix proteins and growth factors altering the properties of ECM and engaging signaling receptors on the cell surface with the activation of latent TGF-beta1. TSP1 promotes aging and it is in turn upregulated by age-related factors as ROS, glucose, and hypoxia and negatively affects mitochondria biogenesis and efficiency as well as endothelial cell proliferation [70,71].

TGF-beta signaling and expression may lead to increased collagens’ deposition and to the development of an age-dependent fibrotic environment correlated to fibrin/fibrinogen accumulation, thus suggesting a possible involvement of macrophages within a generalized inflammatory response [72]. These data support the concept that inflammatory and coagulation pathways are increasingly active with aging and can contribute to sarcopenia and frailty [73].

The matrisome structure and organization are also sustained by the interaction between **proteoglycans** (PGs) and collagens [74]. For instance, biglycan and lumican bind to collagen regulating collagen fibrillogenesis, collagen fibril thickness, and notably interfibrillar spacing, which are important for tissue integrity. Interestingly, in old vs. adult mice, collagen bundles were characterized by the increased distance between fibrils in the presence of similar fibril size. This necessitates further, more detailed analysis of PGs/collagens interactions in aging skeletal muscle. PGs also bind to different growth factors, influencing their bioavailability, cell proliferation, and matrix deposition [75]. In this study, we have observed higher levels of some PGs (e.g., lumican, biglycan, and asporin) in old compared to adult SM. These results are not consistent with those of a recent study, which highlighted a downregulated gene expression of laminin, fibronectin, nidogen, and biglycan in aged SM [55]. This discrepancy is not surprising as many studies have already demonstrated that transcriptomic data had a low correlation with their corresponding proteins [76,77,78]. For instance, the decoupling of the proteome and transcriptome can be due to (i) post-transcriptional mechanisms regulating protein abundance; (ii) greater stability of proteins compared to transcripts; (iii) decreased proteolytic activity; and (iv) changes in protein synthesis rates.

### 3.2. Alteration of Matrisome-Associated Proteins in Aged vs. Adult Mice 

Among **ECM affiliated proteins**, annexins (ANX) can play a critical role in cell membrane repair. The sarcolemma is subjected to severe mechanical stress and continuous stretching and the repair machinery is required to preserve and maintain membrane integrity. ANXA6 is a key factor in the repair machinery, and accumulates at the site of sarcolemma injury. This event precedes the activation of ANXA1 and ANXA2. ANXA6 is the only protein that we have observed to be significantly reduced in aging muscle. Reduction of ANXA6 inhibits the aggregation of ANXA1, ANXA2, and ANXA5, thus negatively affecting the repair process [79]. Moreover, an excess of ANXA2 that leaks from injured myofibers can activate muscle-resident fibro-adipogenic precursors that differentiate into adipocytes which gradually lead to muscle degeneration [80].

Muscle repair processes are also modulated by myoblasts-matrix interactions through interaction between laminin and galectin 1 [81]. Since galectin 1 is upregulated in old SM, these data confirm the higher expression that was detected in the signature of the age-dependent sarcopenia that is involved in mediating cellular responses to inflammation and apoptosis [82], in the terminal differentiation of myoblasts and the disruption of adhesion of myoblasts to laminin [83].

The homeostasis of the matrisome is finely controlled by different **ECM regulators**.

Both in PBS and U/T extract we revealed an increase of cathepsin D in old compared to adult SM. Cathepsin D exhibits its activity in the lysosome, but when the cells break it is released in the extracellular space. This molecule has been recently proposed as a sarcopenia biomarker, as its levels were significantly higher in sarcopenic patients than in control subjects [84].

Similarly, matrix metalloproteases (MMPs), ECM-associated enzymes, play a crucial role in regulating the degradation of matrisome molecules [85]. Except for MMP17, which did not change in aging, other MMPs or their inhibitors have been not identified in this study. This could be due to either their low concentration in our experimental conditions or stringent parameters applied in this study or different resolutions of LC-MS/MS, which is crucial for identifying scarce proteins.

By contrast, the inter alpha trypsin inhibitor heavy chain (ITIH4) is increased in aged SM. However, despite the name, it does not possess intrinsic trypsin inhibitory activity, is upregulated by IL-6, and is involved in the stabilization of the extracellular matrix [86], further supporting the occurrence of molecules involved and favoring the progressive accumulation of ECM in the environment surrounding myofibers.

Moreover, among ECM regulators, we have observed an increase of PEDF, a molecule that, inducing endothelial cell apoptosis, exerts antiangiogenic properties. Reduction of blood vessels may negatively affect nutrient and oxygen availability and therefore muscle metabolism. PEDF is upregulated in the aging muscle and is highly expressed in mesenchymal stem cells from old donors [87].

## 4. Materials and Methods

### 4.1. Reagents

All reagents were purchased from Sigma-Aldrich (Merk KGaA, Taufkirchen, Germany) unless otherwise stated.

### 4.2. Mice

Male BALB/c mice aged 12 months (adult, n = 9) and 24 months (old, n = 9) were used in this study. The mice, housed in groups of three to four, were maintained under standard conditions (24 ± 1 °C ambient temperature, 60 ± 15% relative humidity, and 12 h light/dark cycle) and fed *ad libitum* with standard commercial chow. The animals had only spontaneous free-moving activity in the cage.

### 4.3. Proteomic Analysis

#### 4.3.1. Preparation of Protein Samples for Proteomic Analysis

The whole gastrocnemius muscles were quickly removed from three mice for each age group and froze for proteomic analysis. Each muscle underwent a three-step sequential extraction (Figure 1a). Frozen muscle was homogenized on ice in phosphate buffer (PBS; 1:10 *w*/*v*) using a glass homogenizer to solubilize cellular and hydrophilic proteins. Samples were centrifuged at 8000× *g* for 30 min at 4 °C. After recovering the supernatants (PBS extract), the resulting pellets were resuspended in urea-thiourea buffer (1:10 *w*/*v*) and incubated at 4 °C for 24 h under continuous shaking. After centrifugation at 15,000× *g* for 20 min at 4 °C, the supernatant (U/T extract) was recovered, and the pellet was furthermore homogenized in guanidinium-HCl buffer (pH = 8.5, 1:5 *w*/*v*), heated at 100 °C for 10 min and collected (GuHCl extract).

These last steps were provided to maximize the extracellular matrix protein recovery. The protein concentration of each fraction was quantified using the Bradford method. 200 µg of proteins/fraction/muscle were subjected to proteomic analysis.

For each fraction, a gel-tube digestion was performed as already described [88]. The proteins embedded in the gel-tube were reduced by incubation with 10 mM dithioerythritol/100 mM ammonium bicarbonate for 45 min at 56 °C and then alkylated with 55 mM iodoacetamide/100 mM ammonium bicarbonate for 30 min at RT in the dark. Proteins were digested overnight at 37 °C in 100 mM ammonium bicarbonate pH 8.0 using sequencing grade trypsin at a 1:100 enzyme-to-protein ratio (Promega, Madison, MI, USA). Peptides were extracted from gel-tubes with 100% acetonitrile and dried with a SpeedVac (Eppendorf AG, Hamburg, Germany).

#### 4.3.2. Liquid Chromatography with Tandem Mass Spectrometry (LC-MS/MS)

Peptides obtained from each fraction were resuspended in water/formic acid solution (95:3:2). A UHPLC ultimate 3000 system coupled online to a Q Exactive Hybrid Quadrupole-Orbitrap Mass Spectrometer (Thermo Fisher Scientific, Waltham, MA, USA) was used, as already described [89] with some modifications. Chromatographic separation of peptides took place in a reverse-phase C18 column (100 mm × 2.1 μm ID, 1.9 μm, HyperG Thermo Fischer Scientific) and elution was performed using a binary system of solvents. Solvent A was 0,1% formic acid and solvent B was pure acetonitrile. A linear binary gradient was applied to eluate the peptides: 0–20% solvent B in solvent A for 240 min and further 60 min of 20–40% solvent B in solvent A. The precursor ion detection was done in an m/z-range from 400 to 1800 and the acquisition range for fragment ions was m/z from 200 to 2000. The sample injection flow was 0.5 mL/min, and the column was kept at a constant temperature of 30 °C. Data acquisition was controlled by Xcalibur 2.0.7 Software (Thermo Fisher Scientific, USA). Six independent experiments (three for adult and three for old muscles) were performed and each fraction obtained from each muscle was analyzed in duplicate.

#### 4.3.3. Data Processing for Protein Identification and Quantification

Raw MS/MS data (.raw) were inspected using BatMass (v. 0.3.0). Replicates were aligned using FreeStyle (v.1.5) to check the quality of runs. Raw files were further converted by msConvert ProteoWizard (v.3.0.19239) in MGF file using default settings and uploaded to MASCOT server (v.2.7.0) for MS/MS Ion Search. The search was performed using Uniprot (2018_05) restricted to *Mus musculus* (Taxonomy ID: 10090) and cRAP database to detect most commonly adventitious proteins (https://www.thegpm.org/GPM/index.html (accessed on 6 April 2021:)). Parameters for identification included: (i) trypsin as an enzyme with 1 as maximum missed cleavage; (ii) mass error tolerances for precursor and fragment ions set to 10 ppm and 0.02 Da, respectively; (iii) peptide charge (2+, 3+, 4+); (iv) protein mass no restriction; and (v) carbamidomethyl cysteine (C) was set as fixed modification while deamidation of asparagine and glutamine (NQ), oxidation of methionine (M) and cysteine propionamide (C) were considered as variable modifications. Only confident peptide identified with a false discovery rate ≤ 1 and protein with at least one unique peptide were exported.

Mascot results (.dat) obtained for each extraction were imported in Skyline-daily (v.21.0.9.139) to generate the spectral libraries using the following parameters: 0.95 as spectra cut-off score; peptide length of 8-25 amino acids; precursor ion charge 2+, 3+, 4+; MS1 filters were set to “use high selectivity extraction” with a resolving power of 60,000 at 300 m/z; repeated and duplicate peptides were removed. Accordingly, to Skyline “DDA peptide search” workflow, raw files (.raw) were imported and matched to spectral libraries to recover precursor ion intensity. Precursor ion intensity is the sum of areas under the curve of extracted ion chromatograms (XICs) containing precursor ion isotope peaks (M, M+1, M+2) [90]. Fasta files containing proteins with 1% FDR were imported to Skyline to maintain and fix FDR. Finally, quantitative analysis was performed on proteins with at least two peptides. Indeed, protein inference is mainly based on the unique peptides that are not shared by multiple proteins. However, one peptide can be incorrectly quantified across LC-MS runs, producing improper quantification. In this perspective, first of all, a proteome discovery study was performed releasing a list of proteins identified with at least one unique peptide, and secondly, a label-free quantification study was performed only on proteins identified with at least two peptides to reduce incorrect quantification [37].

### 4.4. Bioinformatic Analysis

We interrogated the MatrisomeDB (http://www.pepchem.org/matrisomedb (accessed on 28 May 2021)) to further characterize and categorize proteins identified in our samples. To date, MatrisomeDB is the most complete database of ECM proteomic data and matrisomal proteins are divided into the “core matrisome” (i.e., glycoproteins, collagens, and proteoglycans) and the “matrisome-associated proteins” (i.e., ECM-affiliated proteins, ECM regulators, and secreted factors) [35].

### 4.5. ECM Component Evaluation by Immunofluorescence

Three adult and three old mice were deeply anesthetized with Tribromoethanol and sacrificed by cervical dislocation. Gastrocnemius muscles were quickly removed and frozen in liquid nitrogen-precooled isopentane. Transversally sectioned 7-μm thick cryosections were incubated with 1% bovine serum albumin, 2% normal goat serum, 0.3% Triton^®^ X-100 in phosphate buffer solution (PBS) for 1 h and immunolabelled with the following probes: a rabbit polyclonal antibody direct against type I collagen, diluted 1:50 (GeneTex, Irvine, CA, USA); a human monoclonal antibody direct against type VI collagen, diluted 1:1000 (ICN Biomedicals, Costa Mesa, CA, USA); rabbit polyclonal antibody direct against laminin, diluted 1:800 (Abcam, Cambridge, UK), in double labeling with anti-collagen type VI.

After washing with PBS, cryosections were stained with the proper secondary antibody: Alexafluor 488-anti-mouse antibodies diluted 1:200 for type VI collagen (Molecular Probes, Invitrogen, CA, USA). Alexafluor 594-anti-rabbit diluted 1:200, for laminin and collagen type I. The cryosections were finally counterstained for DNA with 0.1 μg/mL Hoechst 33258 and mounted in PBS:glycerol (1:1).

An Olympus BX51 microscope equipped with a 100W mercury lamp (Olympus Italia, Milan, Italy) was used under the following conditions: 450–480 nm excitation filter (excf), 500 nm dichroic mirror (dm) for Alexa 488; 540 nm excf, 580 nm dm, and 620 nm barrier filter (bf) for Alexa 594; 330–385 nm excf, 400 nm dm, and 420 nm bf, for Hoechst 33342. Images were recorded with an Olympus Magnifire digital camera system (Olympus Italia).

A routine was written in MATLAB (2018b version, Mathworks) to quantify the area covered by fluorescence-positive pixels for the total area in four random images acquired at X20 for each animal.

### 4.6. Ultrastructural Morphological and Morphometrical Evaluation

Three adult and three old mice were deeply anesthetized with Tribromoethanol drug and then perfused via the ascending aorta with 0.1 M PBS followed by 4% paraformaldehyde in PBS. The gastrocnemius muscle was quickly removed, and samples (about 1 mm^3^) were further placed for 2 h at 4 °C in 2.5% glutaraldehyde (Electron Microscopy Sciences, Hatfield, PA, USA) plus 2% paraformaldehyde in 0.1 M PBS. The samples were then rinsed with PBS, postfixed with 1% OsO_4_ (Electron Microscopy Sciences) and 1.5% potassium ferrocyanide for 2 h at 4 °C, dehydrated with acetone, and embedded in Epon 812 resin (Electron Microscopy Sciences).

Ultrathin sections (70–90 nm thick) were stained with lead citrate for 1 min and observed in a Philips Morgagni transmission electron microscope operating at 80 kV and equipped with a Megaview III camera for digital image acquisition.

The morphometric evaluation of the endomysium thickness was performed on 20 randomly selected electron micrographs (×5600) of longitudinally sectioned muscle, measuring the distance between the sarcolemma of two adjacent muscle cells every 1 µm of sarcolemma length, for a total of 50 measurements per animal.

For morphometric evaluation of the thickness of the basement membrane, a total of 30 measurements per animal were performed. The thickness of the electrodense sheath covering the myofiber was considered on randomly selected electron micrographs (×36,000) of longitudinally sectioned muscles.

The morphometric evaluation of the perimysium collagen bundle size was performed on a total of 10 longitudinally sectioned collagen bundles per animal. The index of collagen bundle linearity (X/Y, expressed as the ratio between the real length of the bundle profile and the corresponding linear length) was assessed on a total of 30 longitudinally sectioned collagen bundles per age group.

For morphometric evaluation of collagen fibrils, measurement of fibril size, as well as the distance between single collagen fibrils, was performed on a total of 100 longitudinally sectioned collagen fibrils per age group.

All measurements were made by using the Radius software for image acquisition and elaboration implemented in the Philips Morgagni transmission electron microscope.

### 4.7. Statistical Analysis

Statistical analysis of proteomic data was performed using Skyline group comparison tool, set as follows: normalization of runs was performed using “Equilize Medians” with 0.95 as the confidence level; Tukey’s Median Polish was set as the summary method. The comparison was set comparing the value obtained in old skeletal muscle against those of adult samples. Proteins with a fold change > 1.5 or < 0.66 and a *p* value < 0.05 were significantly up-and down-regulated, respectively.

Data on the percentage of fluorescent (positive) areas for laminin, collagen type VI, and type I and data of morphometrical evaluation of transmission electron microscopy images for endomysium size, basement membrane thickness, collagen bundle size and linearity, collagen fibril size, and distance were pooled according to the age group and presented as mean ± SEM. The Shapiro–Wilk test showed that data were not normally distributed (*p* < 0.001). Consequently, the subsequent statistical analysis was performed using the non-parametric Mann–Whitney test, setting statistical significance at *p* value ≤ 0.05. The IBM-SPSS (v.25, Armonk, NY, USA) statistical package was used for all analyses.

## 5. Conclusions

This study provided the first characterization of the matrisome in the aging gastrocnemius muscle and highlights the higher age-dependent abundance of several identified ECM components (Figure 6).

Since interactions between ECM molecules and between ECM and sarcolemma provide not only structural support but also a mechano-sensing transduction system and a source of cytokines and growth factors, the deepen insight on the matrisome in the aged SM can pave the way for a better understanding of the synergic interplay of the whole extracellular environment and of the mechanisms that can contribute to the age-dependent muscle dysfunction by hindering, for instance, fiber contractility [91], lateral force transmission [92,93], tissue stiffness [49,50], and satellite cell function [12] through ECM-driven signaling pathways.

## Figures and Tables

**Figure 1 ijms-22-10564-f001:**
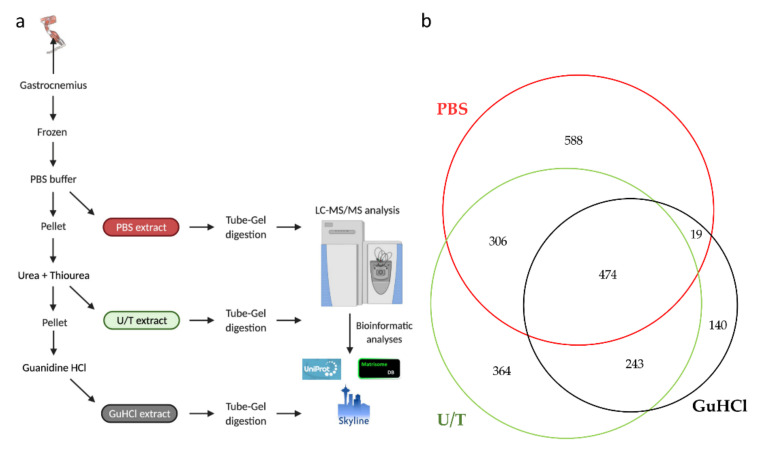
(**a**) Sequential protein extraction from gastrocnemius muscle. (**b**) The Venn diagram shows the number of proteins identified in each extract.

**Figure 2 ijms-22-10564-f002:**
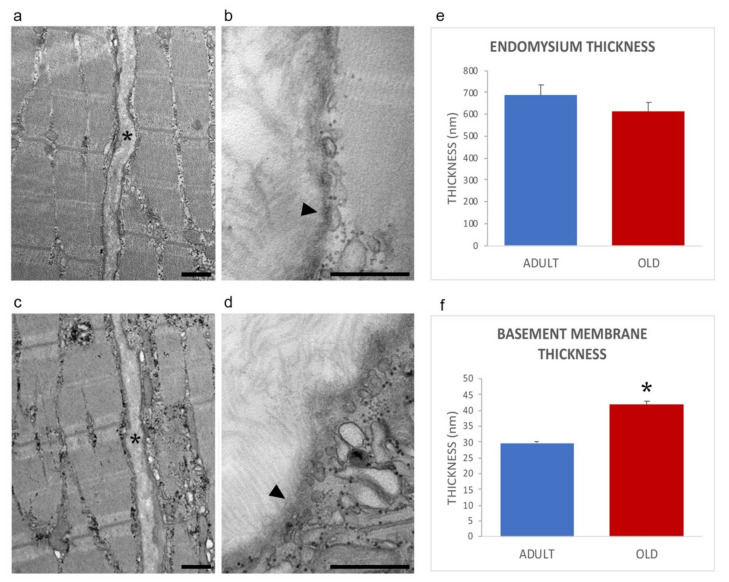
Ultrastructural images of the endomysium in adult (**a**,**b**) and old (**c**,**d**) gastrocnemius muscle. *, endomysium; arrowhead, basement membrane. Bars: 500 nm. The endomysium and basement membrane thickness are reported in panels (**e**) and (**f**), respectively. * *p* < 0.001.

**Figure 3 ijms-22-10564-f003:**
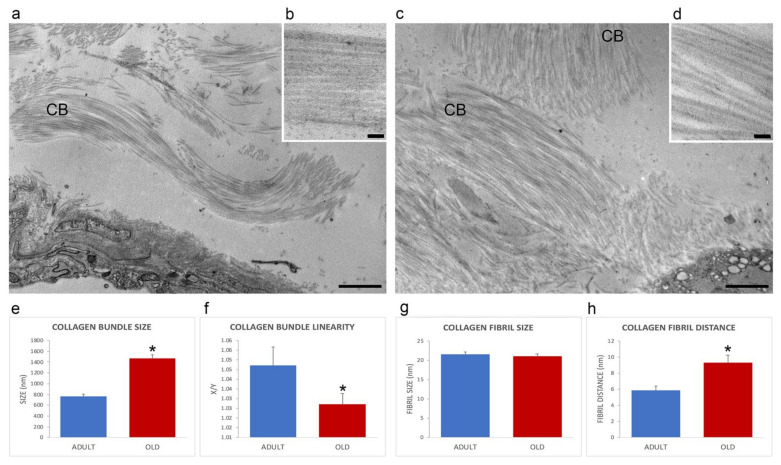
Ultrastructural images (**a**–**d**) of the perimysium of adult (**a**,**b**) and old (**c**,**d**) gastrocnemius muscle. CB, collagen bundle. (**a**,**c**), bars: 1 µm; (**b**,**d**), bars: 100 nm. Morphometric analysis of collagen bundles and collagen fibrils features are reported in the histograms (**e**–**h**). X/Y, ratio between the real length of the collagen bundle profile and the corresponding linear length. * *p* < 0.05.

**Figure 4 ijms-22-10564-f004:**
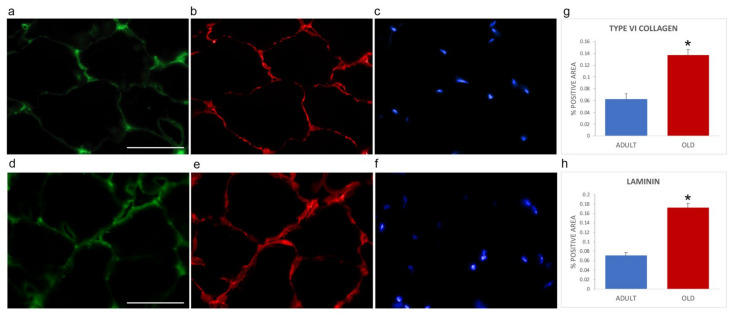
Type VI collagen (**a**,**d**) and laminin (**b**,**e**) immunolabelling of gastrocnemius cryosections in adult (**a**–**c**) and old (**d**–**f**) mice. DNA was counterstained with Hoechst (**c**,**f**). Bar: 50 µm. Panels (**g**) and (**h**), respectively, show quantification of the immunolabelling for type VI collagen and laminin expressed as % of total area). * *p* < 0.001.

**Figure 5 ijms-22-10564-f005:**
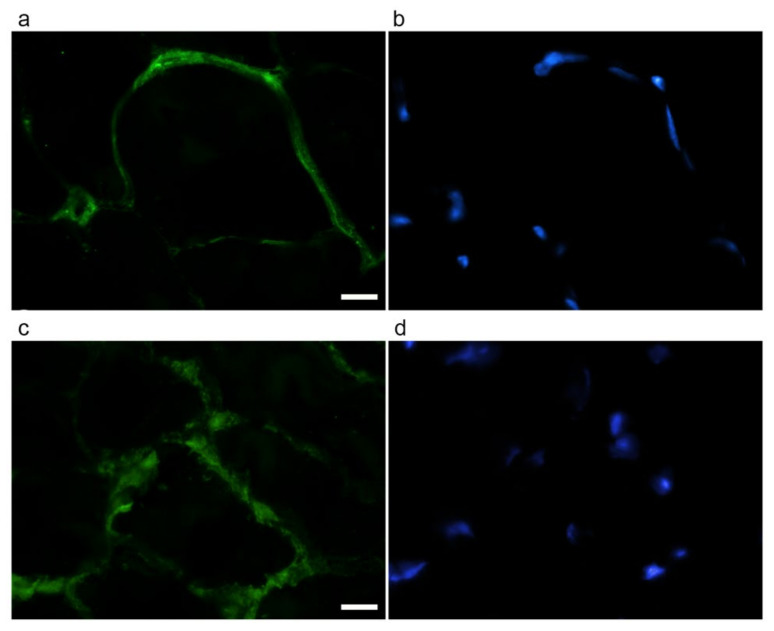
Type I collagen (**a**,**c**) immunolabelling of gastrocnemius cryosections of adult (**a**,**b**) and old (**c**,**d**) mice. DNA was counterstained with Hoechst (**b**,**d**). Bar: 10 µm.

**Figure 6 ijms-22-10564-f006:**
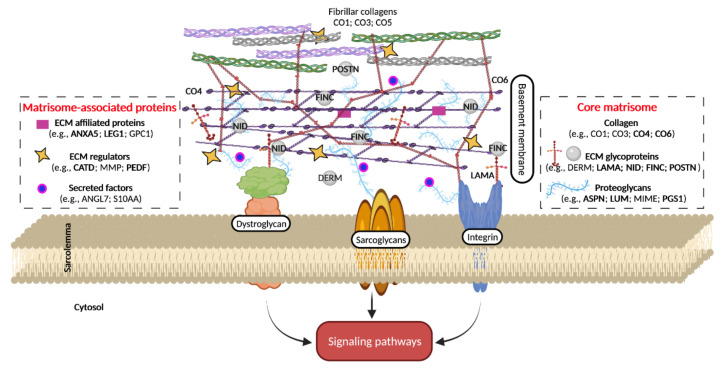
Schematic representation of matrix-sarcolemma axis of skeletal muscle fibre. Proteins differentially expressed between old and adult SM are reported in bold. ANGL7, angiopoietin-related protein 7; ANX, annexin; ASPN, asporin; CO, collagen; CATD, cathepsin D; DERM, dermatopontin; FINC, fibronectin; GPC1, Glypican 1; LAMA, laminin; LEG1, galectin 1; LUM, lumican; MMP, matrix metalloproteinase; MIME, mimecan; NID, nidogen; PEDF, pigment epithelium-derived factor; PGS1, biglycan; POSTN, periostin; S10AA, protein S100-A10.

**Table 1 ijms-22-10564-t001:** ECM and ECM-associated components from the gastrocnemius muscle categorized according to MatrisomeDB.

Protein Symbol	Protein Name	Protein Symbol	Protein Name
**MATRISOME DIVISION:** CORE MATRISOME			
**Category:** collagen			
CO1A1	Collagen alpha-1(I) chain	CO6A1	Collagen alpha-1(VI) chain
CO1A2	Collagen alpha-2(I) chain	CO6A2	Collagen alpha-2(VI) chain
CO3A1	Collagen alpha-1(III) chain	CO6A6	Collagen alpha-6(VI) chain
CO4A1	Collagen alpha-1(IV) chain	COBA1	Collagen alpha-1(XI) chain
CO4A2	Collagen alpha-2(IV) chain	COEA1	Collagen alpha-1(XIV) chain
CO5A1	Collagen alpha-1(V) chain	COFA1	Collagen alpha-1(XV) chain
CO5A2	Collagen alpha-2(V) chain	COIA1	Collagen alpha-1(XVIII) chain
**Category:** ECM-glycoproteins			
ADIPO	Adiponectin	LAMB1	Laminin subunit beta-1
AGRIN	Agrin	LAMB2	Laminin subunit beta-2
BGH3	Transforming growth factor-beta-induced protein ig-h3	LAMC1	Laminin subunit gamma-1
CILP1	Cartilage intermediate layer protein 1	MFAP2	Microfibrillar-associated protein 2
CILP2	Cartilage intermediate layer protein 2	MFAP4	Microfibril-associated glycoprotein 4
COMP	Cartilage oligomeric matrix protein	MFAP5	Microfibrillar-associated protein 5
CREL1	Cysteine-rich with EGF-like domain protein 1	MFGM	Lactadherin
DERM	Dermatopontin	NID1	Nidogen-1
FBN1	Fibrillin-1	NID2	Nidogen-2
FBN2	Fibrillin-2	POSTN	Periostin
FIBA	Fibrinogen alpha chain	SLIT2	Slit homolog 2 protein
FIBB	Fibrinogen beta chain	SRPX	Sushi-repeat-containing protein SRPX
FIBG	Fibrinogen gamma chain	SSPO	SCO-spondin
FINC	Fibronectin	TENA	Tenascin
IGS10	Immunoglobulin superfamily member 10	TINAL	Tubulointerstitial nephritis antigen-like
LAMA2	Laminin subunit alpha-2	TSP1	Thrombospondin-1
LAMA3	Laminin subunit alpha-3	TSP4	Thrombospondin-4
LAMA4	Laminin subunit alpha-4	VMA5A	von Willebrand factor A domain-containing protein 5A
LAMA5	Laminin subunit alpha-5	VWA1	von Willebrand factor A domain-containing protein 1
**Category:** Proteoglycans			
ASPN	Asporin	PGBM	Basement membrane-specific heparan sulfate proteoglycan core protein
CHADL	Chondroadherin-like protein	PGS1	Biglycan
FMOD	Fibromodulin	PGS2	Decorin
LUM	Lumican	PRELP	Prolargin
MIME	Mimecan	PRG2	Bone marrow proteoglycan
**MATRISOME DIVISION:** MATRISOME-ASSOCIATED PROTEINS			
**Category:** ECM affiliated proteins			
ANXA1	Annexin A1	GPC1	Glypican-1
ANXA2	Annexin A2	HEMO	Hemopexin
ANXA3	Annexin A3	LEG1	Galectin-1
ANXA4	Annexin A4	LEGL	Galectin-related protein
ANXA5	Annexin A5	LMAN1	Protein ERGIC-53
ANXA6	Annexin A6	PLXA4	Plexin-A4
ANXA7	Annexin A7	PLXB3	Plexin-B3
ANX11	Annexin A11		
**Category:** ECM regulators			
A1AT1	Alpha-1-antitrypsin 1-1	ITIH1	Inter-alpha-trypsin inhibitor heavy chain H1
A1AT2	Alpha-1-antitrypsin 1-2	ITIH2	Inter-alpha-trypsin inhibitor heavy chain H2
A1AT3	Alpha-1-antitrypsin 1-3	ITIH3	Inter-alpha-trypsin inhibitor heavy chain H3
A1AT4	Alpha-1-antitrypsin 1-4	ITIH4	Inter alpha-trypsin inhibitor, heavy chain 4
A2AP	Alpha-2-antiplasmin	ITIH5	Inter-alpha-trypsin inhibitor heavy chain H5
AMBP	Protein AMBP	KNG1	Kininogen-1
ANGT	Angiotensinogen	MMP17	Matrix metalloproteinase-17
ANT3	Antithrombin-III	PEDF	Pigment epithelium-derived factor
CATB	Cathepsin B	PLMN	Plasminogen
CATD	Cathepsin D	PZP	Pregnancy zone protein
CBG	Corticosteroid-binding globulin	SERPH	Serpin H1
CPN2	Carboxypeptidase N subunit 2	SPA3K	Serine protease inhibitor A3K
CYTB	Cystatin-B	SPA3M	Serine protease inhibitor A3M
FA12	Coagulation factor XII	SPA3N	Serine protease inhibitor A3N
HEP2	Heparin cofactor 2	SPI2	Serpin I2
HRG	Histidine-rich glycoprotein	SULF2	Extracellular sulfatase Sulf-2
HYAL2	Hyaluronidase-2	TGM2	Protein-glutamine gamma-glutamyltransferase 2
IC1	Plasma protease C1 inhibitor	THRB	Prothrombin
ILEUA	Leukocyte elastase inhibitor A		
**Category:** secreted factors			
ANGL7	Angiopoietin-related protein 7	S10A1	Protein S100-A1
FILA2	Filaggrin-2	S10A4	Protein S100-A4
HGFA	Hepatocyte growth factor activator	S10A6	Protein S100-A6
INHBA	Inhibin beta A chain	S10AA	Protein S100-A10
MEG11	Multiple epidermal growth factor-like domains protein 11	WN10A	Protein Wnt-10a

**Table 2 ijms-22-10564-t002:** List of matrisome proteins whose amount statistically significantly changed in old vs. adult gastrocnemius muscles. Quantification was performed by label-free mass spectrometry.

PBS Extract				
Category	Protein Symbol	Protein Name	Ratio (Old/Adult) Log_2_ Fold Change	*p*-Value
ECM regulators	A1AT2	Alpha-1-antitrypsin 1-2	0.69	0.006
	A2AP	Alpha-2-antiplasmin	1.04	0.038
	CATD	Cathepsin D	0.58	0.001
	CBG	Corticosteroid-binding globulin	1.82	0.049
	ILEUA	Leukocyte elastase inhibitor A	1.11	0.008
	ITIH2	Inter-alpha-trypsin inhibitor heavy chain H2	2.00	0.038
	ITIH4	Inter alpha-trypsin inhibitor, heavy chain 4	1.42	0.018
	KNG1	Kininogen-1	0.97	0.043
	PEDF	Pigment epithelium-derived factor	1.11	0.040
	PZP	Pregnancy zone protein	0.86	0.012
	SPA3N	Serine protease inhibitor A3N	1.77	0.028
ECM glycoproteins	FIBA	Fibrinogen alpha chain	1.19	0.001
	FIBB	Fibrinogen beta chain	1.19	0.001
	FIBG	Fibrinogen gamma chain	1.14	0.001
	TSP4	Thrombospondin-4	1.44	0.000
	VMA5A	von Willebrand factor A domain-containing protein 5A	0.81	0.000
Proteoglycans	ASPN	Asporin	0.53	0.004
	LUM	Lumican	0.48	0.030
	PRELP	Prolargin	0.37	0.001
ECM affiliated proteins	ANXA4	Annexin A4	0.63	0.002
	ANXA5	Annexin A5	0.51	0.009
	ANXA6	Annexin A6	−0.39	0.008
	ANX11	Annexin A11	0.28	0.021
	HEMO	Hemopexin	0.74	0.018
**U/T Extract**				
Collagen	CO4A1	Collagen alpha-1(IV) chain	1.30	0.016
	CO6A1	Collagen alpha-1(VI) chain	0.61	0.000
	CO6A2	Collagen alpha-2(VI) chain	0.59	0.001
ECM regulators	A1AT2	Alpha-1-antitrypsin 1-2	0.87	0.022
	CATD	Cathepsin D	1.04	0.000
	TGM2	Protein-glutamine gamma-glutamyltransferase 2	0.80	0.013
ECM glycoproteins	BGH3	Transforming growth factor-beta-induced protein ig-h3	0.78	0.016
	CILP1	Cartilage intermediate layer protein 1	1.47	0.000
	DERM	Dermatopontin	0.51	0.004
	FINC	Fibronectin	1.35	0.036
	LAMA2	Laminin subunit alpha-2	0.45	0.000
	LAMA5	Laminin subunit alpha-5	0.79	0.002
	LAMB1	Laminin subunit beta-1	0.47	0.010
	LAMB2	Laminin subunit beta-2	0.53	0.000
	LAMC1	Laminin subunit gamma-1	0.56	0.000
	NID1	Nidogen-1	0.54	0.000
	NID2	Nidogen-2	1.09	0.000
	POSTN	Periostin	1.28	0.003
	TENA	Tenascin	2.55	0.022
	TSP4	Thrombospondin-4	1.34	0.048
Proteoglycans	ASPN	Asporin	0.79	0.008
	FMOD	Fibromodulin	0.73	0.005
	LUM	Lumican	0.81	0.001
	MIME	Mimecan	0.54	0.008
	PGS1	Biglycan	1.05	0.002
	PRELP	Prolargin	0.87	0.007
ECM affiliated proteins	ANXA2	Annexin A2	0.42	0.005
	ANXA5	Annexin A5	2.28	0.002
	HEMO	Hemopexin	1.59	0.007
	ANX11	Annexin A11	0.90	0.000
	LEG1	Galectin-1	1.19	0.021
**GuHCl Extract**				
Collagens	COFA1	Collagen alpha-1(XV) chain	0.67	0.019
ECM glycoproteins	LAMA5	Laminin subunit alpha-5	0.81	0.006
ECM affiliated proteins	ANXA2	Annexin A2	0.67	0.046

## Data Availability

The mass spectrometry proteomics data have been deposited to the ProteomeXchange Consortium via the PRIDE partner repository with the dataset identifier PXD027895.

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
