# Peer review of "Age-Related Changes in the Matrisome of the Mouse Skeletal Muscle"

_ijms, 2021, doi:10.3390/ijms221910564_

Round 1
Reviewer 1 Report
The work by Lofaro and colleagues is a very elegant study concerning the matrisome properties of adult and old mice. The authors observed an increased amount of ECM core- and related- proteins through LC-MS in old vs adult mice matrisome, accompanied by increased basal lamina thickness and perimysium collagen bundle size and linearity as well as collagen 4 and laminin abundance. I thank the authors for this interesting study, confirming that the increased stiffness observed in the aging muscle ECM is potentially related to an increased production and deposition of collagen and ECM-related proteins.
Here below I attach my comments for the authors.
Major comments:
1) The concept of sarcopenia.
- L35: from my knowledge, and from the latest guidelines (Cruz-Jentoft et al., 2019) sarcopenia is not defined as just a "progressive decrease of muscle mass and strength associated with changes in tissue composition and impaired neuromuscular driving forces" occurring in healthy aging, as stated by the authors. In the latest guidelines, indeed, sarcopenia is defined as a multifactorial pathological condition, now identified as low muscle strength (below a certain cut-off) accompanied by low muscle quality or quantity. Therefore, until a certain amount, healthy aging-associated loss of muscle mass, force and motoneurons does not automatically mean sarcopenia is developed.
I suggest the authors to rephrase the sentence, for example by stating that "the progressive loss of muscle force, muscle mass, alterations in tissue composition and increasing denervation can lead to the development of a pathological condition termed sarcopenia".
- L212: again, SM atrophy and myofibre loss does not necessary underlie sarcopenia. Please amend as suggested for the introduction.
2) Results, L150 (Table 2): the shown p-value is an adjusted or unadjusted value?
3) Discussion: overall, the discussion is quite long but nice. However, the authors shift from discussing the potential role/results of the up-regulation of one protein to another without connecting them clearly. In this sense, it could be useful to divide the discussion in sub-paragraphs, for example "Alterations of ECM-core proteins in aged vs adult mice" (or similar), where discussing collagen, glycoproteins and proteoglycans (if you choose to write in bold these words, please do it for all of them, as glycoproteins is not bold in this case); "Alterations of matrisome-associated proteins in aged vs adult mice" (or similar) when discussing the ECM-affiliated proteins and the regulators.
4) The choice to use adult mice instead of young: I understand the rationale behind this choice. However, I feel this study is not complete as it does not consider young mice; although the major alterations will definitely happen between adult and old age, I believe it would be more correct to consider also young. I understand this would mean an incredible amount of work for the authors; thus I am not requiring this analysis to be added to the study, but I require the authors to state this as a limitation of the work.
5) Statistics: was the normality of the dataset assessed? If so, please state it. If not, please test it, through visual inspection of Q-Q plots and specific tests (such as Shapiro Wilk). If normality is not passed, parametric statistics should not be used. Instead, non-parametric tests (such as Mann-Whitney test) should be applied. If the results are different, please correct the work accordingly.
Minor comments:
- Abstract: I do understand that you refer to sarcopenia as the progressive muscle mass, force and function causing frailty. However, sarcopenia is to date defined by specific cut offs of muscle function, thus I believe referring generically to "sarcopenia" as "loss of muscle mass and force" is not correct. I suggest to remove the words (i.e. sarcopenia) or to better explain the concept (see "major comment" 1).
- The introduction has a nice logical flow and reads well; however, there are some very long sentences that could be potentially divided or re-written in order to be more "reader friendly". Specifically, L37-43, L51-56 and L96-101.
- L256: I would definitely not say that "Collagen type IV accumulation has been shown to drive out SC from their niche, thus explaining the satellite cell reduction observed in aging"; I agree this could play a role, but I do not believe this would be the only mechanisms explaining SC depleting in the aging skeletal muscle. Indeed, this has been reported to be related to extrinsic changes in the SM niche (where collagen 4 depletion could play a role) but also by intrinsic changes in SC (Blau et al., 2015). Thus, I suggest the authors to rephrase this sentence, for example by stating that " Collagen type IV accumulation has been shown to drive out SC from their niche, thus potentially contributing to the satellite cell reduction observed in aging".
L 285: This has been also observed in a study recently published on this journal, where increased stiffness of human muscle fibre bundles was observed in aged vs young; this was found to be linked to increased collagen abundance (Pavan et al., 2020)
Bibliography
Blau H, Cosgrove BD & Andrew TVH (2015). The central role of muscle stem cells in regenerative failure with aging. Nat Med 21, 854–862.
Cruz-Jentoft AJ et al. (2019). Sarcopenia: Revised European consensus on definition and diagnosis. Age Ageing 48, 16–31.
Pavan P, Monti E, Bondí M, Fan C, Stecco C, Narici M, Reggiani C & Marcucci L (2020). Alterations of extracellular matrix mechanical properties contribute to age-related functional impairment of human skeletal muscles. Int J Mol Sci 21, 1–14.
Reviewer 2 Report
- Did authors took the whole Gastrocnemius muscle for analysis? The muscle contains the lateral (red) and intermediate (white) parts and there may be some differences in measured parameters in these two morphologically different parts of the muscle (especially the proteome). May authors comment on this issue in the discussion?
- What decided for the choice of collagen type VI, I and laminin only to be investigated by IF? It should be stated in the section 2.4.
- It is interesting that authors did not detect any MMPs despite their role in the ECM healing/remodelling (i.e. as stated in the paper: 10.4161/cam.3.4.9338). I think that Western blot analysis of MMP-1, MMP-2 and MMP-9 would be of the great advantage and add merit to the age-related SM ECM remodelling shown by the authors and based on proteomic analysis; especially I would expect age-related changes in MMP-1 based on the literature (i.e. 10.1016/j.actbio.2008.03.010)
- The number of ethical statement or the information that the study did not require any is missing from the Methods section
